# Calcareous Nannofossil Biostratigraphy and Biochronology at ODP Site 1123 (Offshore New Zealand): A Reference Section for the Last 20 Myr in the Southern Ocean

Agata Di Stefano, Natale Maria D'Andrea [ID], Salvatore Distefano [ID], Salvatore Urso [ID], Laura Borzì [ID], Niccolò Baldassini and Viviana Barbagallo *[ID]

Dipartimento di Scienze Biologiche, Geologiche ed Ambientali, Università degli Studi di Catania, Corso Italia, 57, 95129 Catania, Italy
* Correspondence: viviana.barbagallo@phd.unict.it; Tel.: +39-09-5719-5724

**Abstract:** The quantitative analysis of the calcareous nannofossil content yield in the 600 m thick succession drilled at ODP Site 1123 (offshore New Zealand), considered as a reference section for the Southern Ocean region, allowed the recognition of 43 bioevents distributed along the last 20 Myr. The correlation with the excellent magnetostratigraphic record resulted in the attribution of numerical ages for the position of the detected horizons. Many of the marker species used in previous zonation were detected also at ODP Site 1123, but others revealed to be absent or of scarce applicability. On the other hand, the good applicability of additional events was verified and proved to be useful for the biostratigraphic subdivision and correlation. The obtained average bio- and chronostratigraphic resolution is about 0.6 Myr along the whole section, which increases to about 0.3 in the Pliocene–Holocene time interval. The final result is a detailed southern mid-to-high latitude nannofossil biochronology for the last 20 Myr, which confirms that the ODP Site 1123 succession represents a reference section for the Southern Ocean.

**Keywords:** New-Zealand offshore; nannofossil biostratigraphy and biochronology; Neogene–Quaternary

## 1. Introduction

In 1998, ODP Site 1123 was drilled 410 km NE of the Chatham Islands (offshore New Zealand) at a depth of 3290 m [1] (Figure 1). It was drilled in order to document the stratigraphy of the northern slopes of the Chatam Rise and to establish the effect of the Deep Western Boundary Current on the Neogene sediment deposition. The succession is about 600 m thick and consists of almost exclusively nano ooze with numerous tephra levels in the upper part. The Marshall Paraconformity, e.g., [2,3], represents a major discontinuity that stops the continuity of the succession and connects Lower Miocene to basal Oligocene deposits, with about 12.5 Myr missing.

Several studies were carried out in recent decades on the ODP Site 1123 succession, which made it possible to reconstruct the paleoceanographic and paleoclimatic history of the Southern Ocean region, especially in the Plio-Pleistocene time interval, e.g., [4–10]. Particular attention was paid to the evolution of the Deep Western Boundary Current within the complex framework of deep ocean circulation in specific time intervals, e.g., [11–15].

Other studies focused on the bio- and chronostratigraphic features of the selected time interval, according to different groups of fossils, e.g., [16–18], but at present a detailed biostratigraphic reconstruction of the entire ODP Site 1123 succession is not available.

Calcareous nannofossils are remnants of calcareous unicellular brown algae widely used for the stratigraphic correlation of marine successions. The on-board preliminary analysis revealed that the calcareous nannofossil assemblages were abundant and generally well preserved in almost all the samples. It was soon clear that the ODP Site 1123 succession represented the most complete sedimentary record currently available for the

Lower Miocene–Holocene interval in the study area, well subdivided with a relevant number of biostratigraphic events and the succession also accompanied by an excellent magnetostratigraphic record.

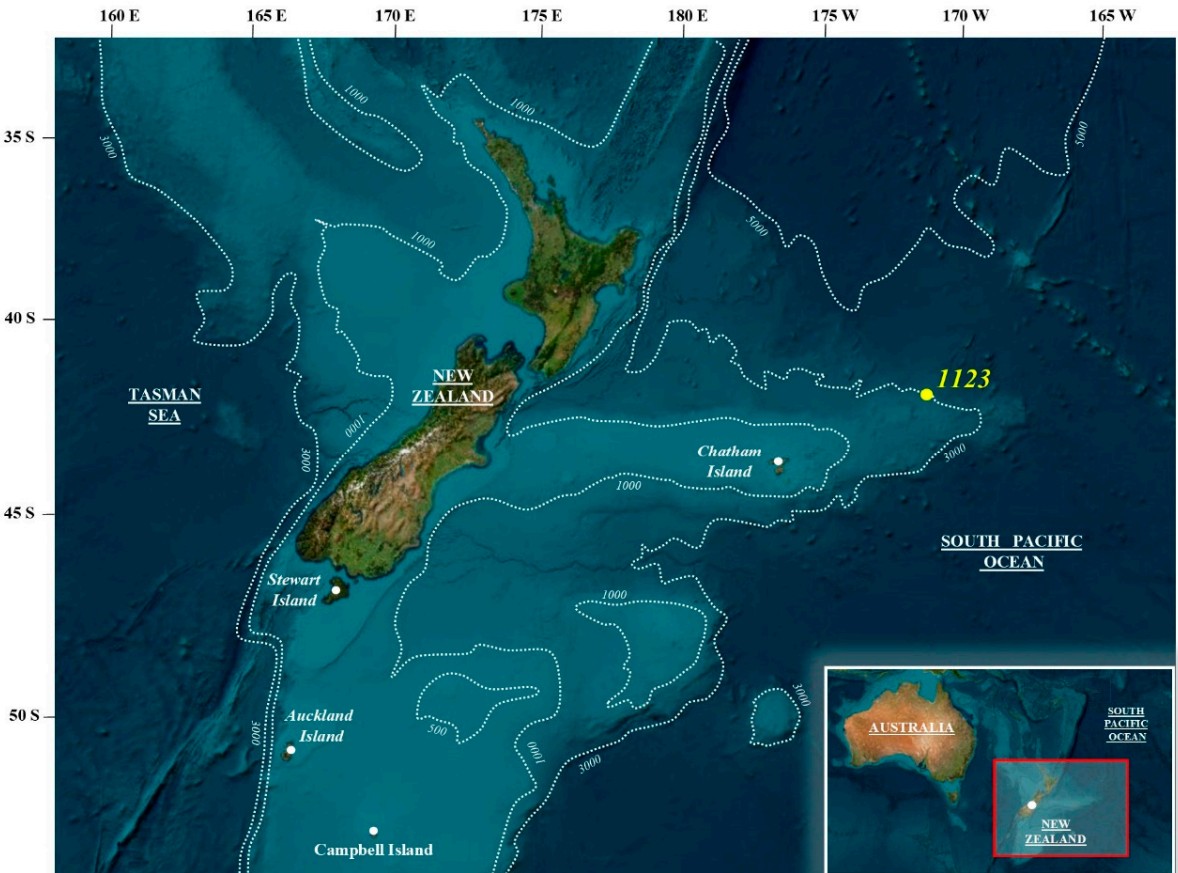

**Figure 1.** ODP Site 1123 Location map.

In this paper, we present the results of a quantitative study of the nannofossil assemblages of the ODP Site 1123 succession, which allowed us to identify 43 bioevents, significantly improving the results obtained from the on-board analysis, where a total of 22 events were detected [1]. The excellent magnetostratigraphic data [1] were recalibrated according to the ATS2020 [18] and integrated with the updated nannofossil biostratigraphy; as such, the age calibration of the recognized bioevents comes out in terms of numerical ages (Ma).

The final result was a detailed southern mid-to-high latitude nannofossil biochronology for the last 20 Myr, which confirmed that the ODP Site 1123 succession represents a reference section for the Southern Ocean.

## 2. Materials and Methods

Cores from ODP Site 1123 recovered a succession of nannofossil ooze, chalk and limestone [1]. The first 250 m were assigned to litostratigraphic Unit I and consisted of light gray clayey nannofossil ooze alternating with white nannofossil ooze, with abundant tephra layers, especially in the upper part. The subsequent 200 m were assigned to the lithostratigraphic Unit II, which was made of homogeneous light greenish gray nannofossil chalk. The lowermost part above the Marshall Paraconformity was about 130 m thick and was assigned to Unit III, and this part consisted of nannofossil mudstone, nannofossil chalk and clay-bearing nannofossil chalk; an interval of about 8 m of thickness was characterized

by plastically deformed clayey nannofossil ooze and reported as "debris flow", and it was present within this unit (Figure 2).

Samples from the nannofossil analysis were collected from cores 1123B-H-1 (H =HPC= Hydraulic Piston Core) to 1123B-H-52 and from 1123C-X-18 (X=XCB=Extended Core Barrel) to 1123C-X-29. Depths of samples were reported in revised meters of the composite depth (mcd) [1]. A total of 368 samples were examined in selected intervals, in addition to those examined onboard by one of the authors of the present paper (ADS). Samples were thus collected at an average distance of 75 cm, which guaranteed the maximum error of 35 cm for the detected bioevents. Standard smear slides [19] were made for all samples using an optical adhesive as a mounting medium and dried under an UV lamp. Calcareous nannofossils were examined by means of a light-polarized microscope at 1000–1600× magnification.

The taxonomy of most of the taxa considered in the present paper were referenced in [19–24]. Useful information is also available from the Nannotax3 website [25].

Quantitative methods were adopted to make one bioevent easily correlatable among different successions and to better highlight significant variations in the calcareous nanno-fossil assemblages [26–28].

In detail, bioevents based on species belonging to the *Calcidiscus*, *Discoaster*, *Gephyro-capsa*, *Helicosphaera* and *Sphenolithus* genera were estimated within a prefixed number of specimens of the same genus, while the other bioevents were evaluated within a minimum of 500 nannofossils > 4 μm in size.

The definitions of the bioevents detected in the present study (Figure 3) basically follow those proposed by Backman et al., 2012 [29], which are: lowest and highest occurrences of marker species, referred to as base (B) and top (T), respectively; levels in which marker species begin to be common or to decline, referred to as Base common (Bc) or Top common (Tc), respectively. Other biohorizons regarded as relevant are: the base and the top of the paracme (interval of the temporary absence of a taxon), referred to as PB and PT, respectively; the base and the top of the acme (the interval of sharp increase in the abundance of a taxon) referred to as AB and AT, respectively; the level where the replacement of a taxon with another occurs it was indicated as the abundance crossover (X).

The chronostratigraphic framework referred to Raffi et al., 2020 [18], for the Neogene Period and to Pillans and Gibbard, 2012 [30], for the Quaternary, and compared with the New Zealand Geological Timescale [31] (Figure 4).

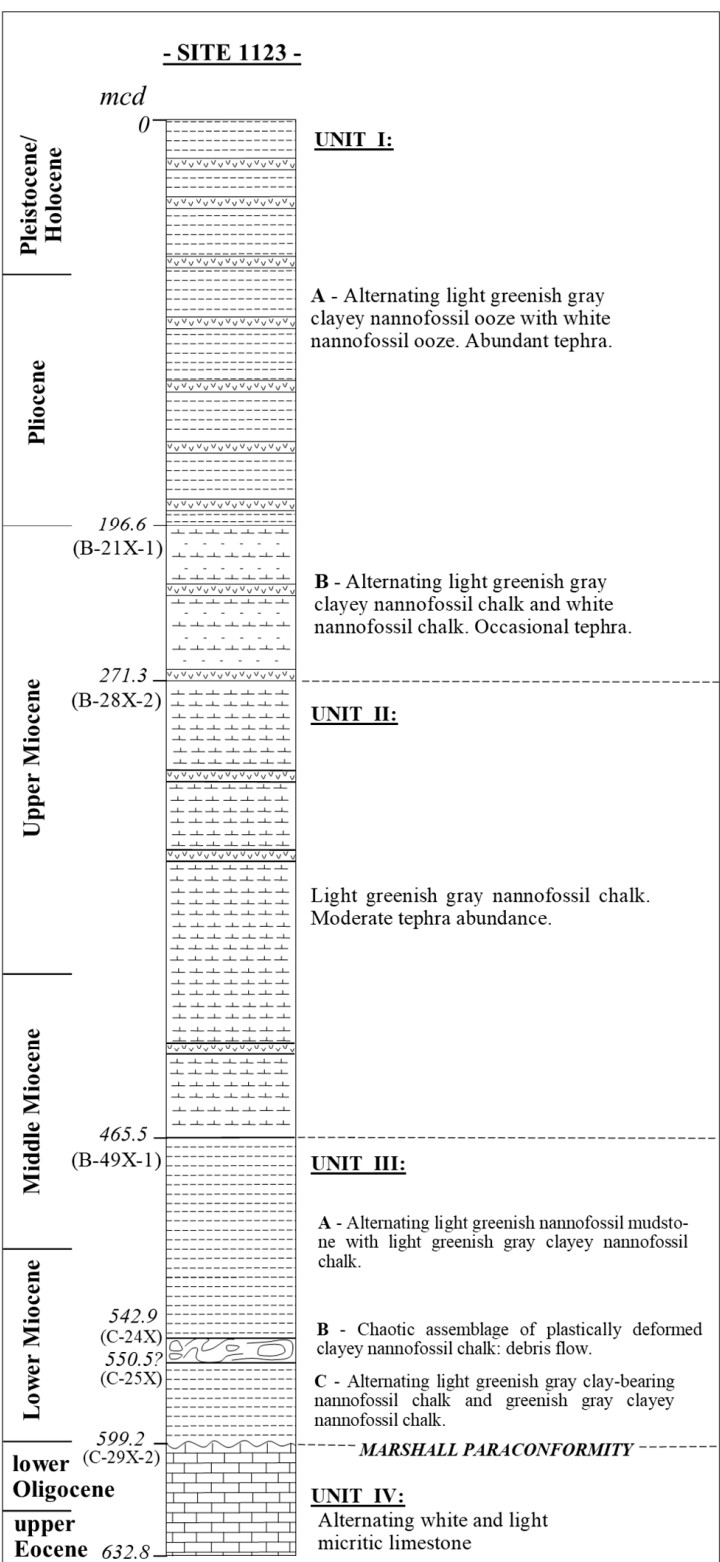

**Figure 2.** Lithostratigraphic characters of ODP Site 1123 succession.

| bioevent | marker species | zonal boundaries | zonal/subzonal boundaries | zonal boundaries | ps |
|---|---|---|---|---|---|
| B | *Emiliania huxleyi* | NN20/NN21 | CN14b/CN15 | | X |
| T | *Pseudoemiliania lacunosa* | NN19/NN20 | | CNMPL10/CNPL11 | X |
| PT | medium *Gephyrocapsa* | | | CNMPL9/CNPL10 | X |
| B | *Gephyrocapsa omega* | | | | X |
| T | *Helicospahera sellii* | | | | X |
| T | *Calcidiscus macintyrei* | | | | X |
| PB | medium *Gephyrocapsa* | | | | X |
| T | large *Gephyrocapsa* | | | CNMPL8/CNPL9 | X |
| B | large *Gephyrocapsa* | | | | X |
| B | medium *Gephyrocapsa* | | | CNMPL7/CNPL8 | X |
| B | *Gephyrocapsa carribeanica* | | CN13a/CN13b | | nd |
| T | *Discoaster brouweri* | NN18/NN19 | CN12d/CN13a | CNMPL6/CNPL7 | X |
| T | *Discoaster pentaradiatus* | NN17/NN18 | CN12c/CN12d | CNMPL5/CNPL6 | X |
| T | *Discoaster surculus* | NN16/NN17 | CN12b/CN12c | | X |
| T | *Discoaster tamalis* | | CN12a/CN12b | CNMPL4/CNPL5 | X |
| T | *Sphenolithus* spp. | | CN11b/CN12a | | nd |
| T | *Reticulofenestra pseudoumbilicus* | NN15/NN16 | CN11b/CN12a | CNMPL3/CNPL4 | X |
| B/Bc | *Discoaster asymmetricus* | NN13/NN14 | CN11a/CN11b | CNMPL2/CNPL3 | X |
| B | *Pseudoemiliania lacunosa* | | | | X |
| Bc | *Helicoshaera sellii* | | | | X |
| T | *Amaurolithus primus + A. tricorniculatus* | | CN10c/CN11a | | nd |
| T | *Ceratholithus acutus* | | CN10b/CN10c | CNMPL1/CNPL2 | nd |
| B | *Ceratholithus rugosus* | NN12/NN13 | CN10b/CN10c | | nd |
| B | *Ceratholithus acutus* | | CN10a/CN10b | CNM20/CNPL1 | nd |
| T | *Discoaster quinqueramus* | NN11/NN12 | CN9b/CN10a | CNM19/CNM20 | X |
| T | *Nicklithus amplificus* | | | CNM18/CNM19 | X |
| B | *Nicklithus amplificus* | | CN9a/CN9b | CNM17/CNM18 | X |
| B | *Amaurolithus delicatus* | | | | X |
| B | *Amaurolithus primus* | | | CNM16/CNM17 | X |
| T | *Minylitha convallis* | | | | X |
| B | *Discoaster berggrenii* | NN10/NN11 | CN8b/CN9a | CNM15/CNM16 | nd |
| B | *Discoaster quinqueramus* | | | | X |
| B | *Discoaster loeblichii + D. neorectus* | | CN8a/CN8b | | nd |
| PB | *Reticulofenestra pseudoumbilicus* | | | CNM14/CNM15 | nd |
| Bc | *Discaoster pentaradiatus* | | | | X |
| T | *Discoaster hamatus* | NN9/NN10 | CN7b/CN8a | CNM13/CNM14 | nd |
| T | *Catinaster coalitus* | | | | X |
| B | *Catinaster calyculus* | | CN7a/CN7b | | nd |
| B | *Minylitha convallis* | | | | X |
| B | *Discoaster hamatus* | NN8/NN9 | CN6/CN7a | CNM12/CNM13 | nd |
| Bc | *Discoaster bellus* | | | | X |
| B | *Catinaster coalitus* | NN7/NN8 | CN5b/CN6 | CNM11/CNM12 | X |
| Tc | *Coccolithus miopelagicus* | | | | X |
| Tc | *Discoaster kugleri* | | | CNM10/CNM11 | X |
| B | *Discoaster kugleri* | NN6/NN7 | CN5a/CN5b | CNM9/CNM10 | X |
| Tc | *Calcidiscus premacintyrei* | | | CNM8/CNM9 | X |
| T | *Sphenolithus heteromorphus* | NN5/NN6 | CN4/CN5 | CNM7/CNM8 | X |
| T | *Helicosphara ampliaperta* | NN4/NN5 | CN3/CN4 | | X |
| X | *Discoaster deflandrei/D. variabilis* | | | | X |
| B | *Discoaster signus* | | | CNM6/CNM7 | nd |
| Tc | *Sphenolithus disbelemnos* | | | | X |
| B | *Sphenolithus heteromorphus* | | CN2/CN3 | CNM5/CNM6 | X |
| T | *Sphenolithus belemnos* | NN3/NN4 | | | X (Tc) |
| B | *Sphenolithus belemnos* | | CN1c/CN2 | CNM4/CNM5 | X (Bc) |
| T | *Triquetrorhabdulus carinatus* | NN2/NN3 | | | X |
| AT | *Cyclicargolithus abisectus (> 10 µm)* | | | | X |

**Figure 3.** List of nannofossil bioevents detected at ODP Site 1123 (X) and bioevents adopted as zonal boundaries in standard [15,18,19,32,33] and most recent zonal schemes for oceanic successions [29]; ps stands for *present study,* nd stands for *not detected.* Most of the nannofossils index markers are illustrated in Figures A1–A3 in Appendix A.

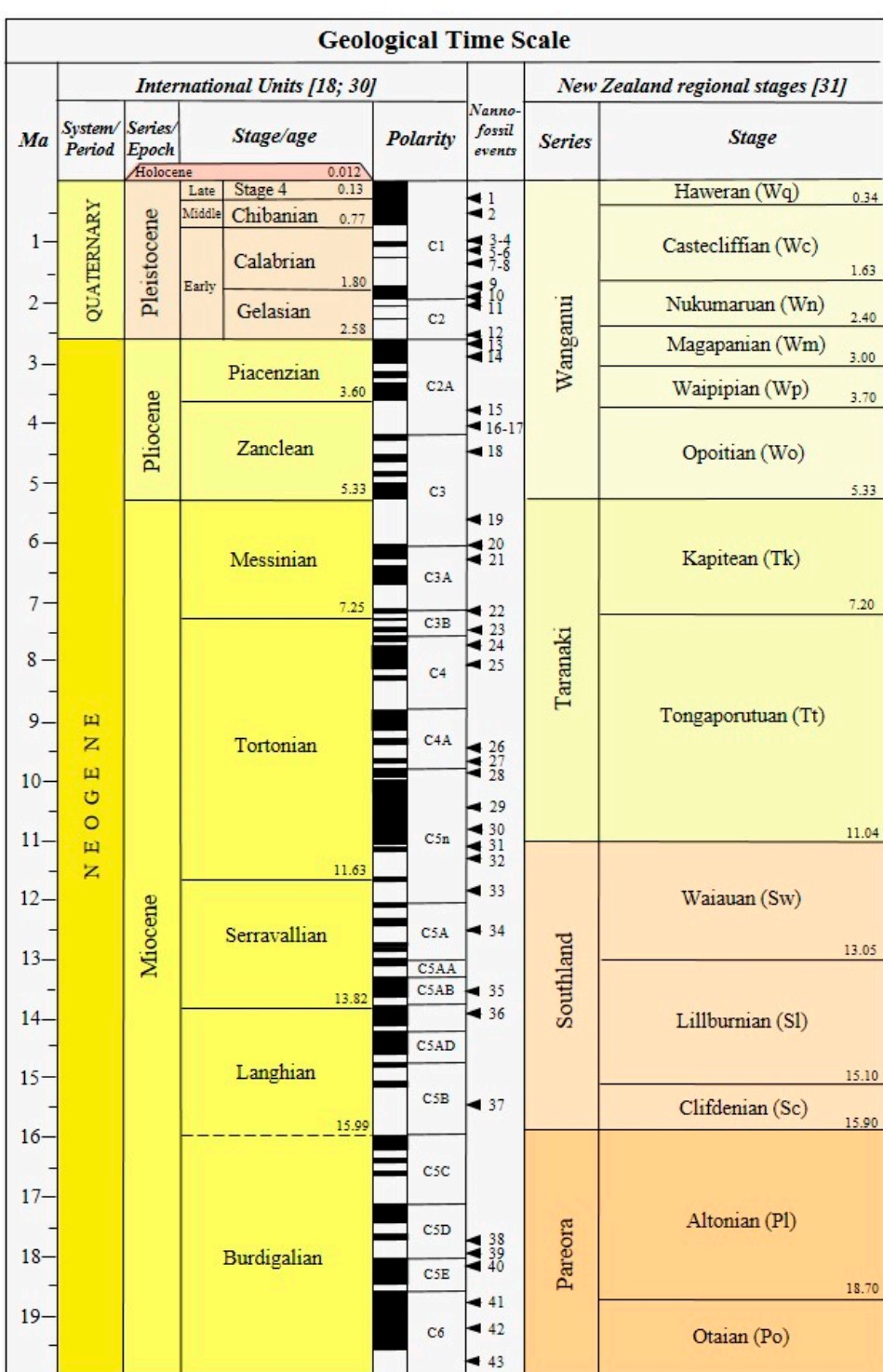

**Figure 4.** Chronostratigraphic framework adopted in the present study. For the International units stages [18,30]; for the New Zealand regional stages [31].

## 3. Neogene–Quaternary Calcareous Nannofossils Biostratigraphy: State of the Art

The biostratigraphic usefulness of calcareous nannofossils, in the Neogene–Quaternary interval, was highlighted in the late 1950s. Pioneering studies are those of Bramlette and Riedel (1954) [34] and subsequently of Bramlette and Wilcoxon, 1967 [35], that represent the basis that gave rise to the standard zonal scheme by Martini, 1971 [32]. Later, the advent of the deep-sea drilling and the recovery of deep-sea sediments led to an ever-increasing improvement in nannofossil biostratigraphic schemes, from Okada and Bukry,

1980 [33] and to Backman et al., 2012 [29] (Figure 3), and this topic field also benefited from advances in other fields of Earth Sciences (e.g., magnetostratigraphy, radiometric dating, cyclostratigraphy, etc.), highlighting the high degree of bio- and chronostratigraphic resolution that this group of organisms can provide in dating sediments of marine origin.

In the following sections, we summarize the main nannofossil biostratigraphic events characterizing the last 20 million years.

### 3.1. Holocene–Pleistocene (Holocene–Gelasian = Haweran–Nukumaruan)

During the Holocene–Pleistocene time interval, gephyrocapsids are important components of the nannofossil assemblages, indispensable for the biostratigraphic subdivision and correlation of marine successions. In the present study, we adopted the following biometrically based definitions of gephyrocapsids [27,36,37], splitting the group into four categories: (i) *Gephyrocapsa* specimens with long axis < 4μm, reported as "small *Gephyrocapsa*"; (ii) *Gephyrocapsa* specimens with long axis ≥ 4μm and open central area (e.g., *G. oceanica* s.l. [27]; not including *G. carribeanica*), reported as "medium *Gephyrocapsa*"; (iii) *Gephyrocapsa* specimens with long axis > 5.5 μm (including *G. carribeanica*), reported as "large *Gephyrocapsa*" and (iv) *Gephyrocapsa* specimens with long axis ≥ 4μm, open central area and a bridge nearly aligned with the short axis, referred to as *Gephyrocapsa omega* (corresponding to *Gephyrocapsa* sp. 3 of [36]). In the Late Pleistocene, relevant bioevents are the first occurrence of *E. huxleyi* and the disappearance of *P. lacunosa*.

### 3.2. Pliocene–Late Miocene (Piacenzian–Tortonian = Mangapanian–Tongaporutuan)

In this time interval, the *Discoaster* genus provides the highest number of bioevents, even if the distribution of Ceratolitidae is also noteworthy in the Zanclean and Messinian Stages. They are represented by the *Amaurolithus*, *Nicklithus* and *Ceratolithus* genera, which, however, can be characterized by a very discontinuous presence. Other relevant bioevents are the disappearance of *R. pseudoumbilicus* and *Sphenolithus* spp.

### 3.3. Middle–Early Miocene (Serravallian–Burdigalian = Waiauan–Otaian)

In the Middle–Early Miocene, many useful biostratigraphic events are based on species belonging to the *Sphenolithus* genus (namely *S. belemnos*; *S. disbelemnos*; *S. heteromorphus*) that are generally well represented in the nannofossil assemblages. Remarkable bioevents in this time interval are represented by the last occurrence (T) of *T. carinatus* and by the first occurrence (B) of *H. ampliaperta*, even if the two species are often discontinuously distributed in oceanic sediments.

The *Discoaster* genus also provides useful bioevents, but the low degree of preservation can in some cases prevent the correct identification of some species, such as *Discoaster exilis*, which due to recrystallization problems can easily be confused with *D. variabilis*.

## 4. Calcareous Nannofossils Biostratigraphy and Biochronology at ODP Site 1123

Nannofossils are generally abundant throughout the sequence; the preservation is good, especially in the top part of the succession, down to about 200 mcd, where dissolution or recrystallization phenomena may be present. Reworking may occur throughout the sequence, in low frequencies.

The study of nannofossil associations at ODP Site 1123 preliminarily focused on recognizing the biostratigraphic events identified in the schemes of [32,33]. Subsequently, the possibility of comparing the data with the more recent zonal scheme of [29] allowed for a more detailed biostratigraphic subdivision of the succession.

Age estimates for the nannofossil bioevents were calculated according to the age model reconstructed for ODP Site 1123. The age model was based primarily on the magnetostratigraphic record, virtually complete from Chron C1n to Chron C6r (Figures 5 and 7), with the correlation of the individual polarity chrons to the geomagnetic polarity time scale being based on shipboard biostratigraphical data from calcareous nannofossils, planktic

foraminifera, radiolaria and the identification of the magnetochron reversal pattern [1]. All magnetostratigraphic reversal ages were recalibrated according to the ATS2020 [18].

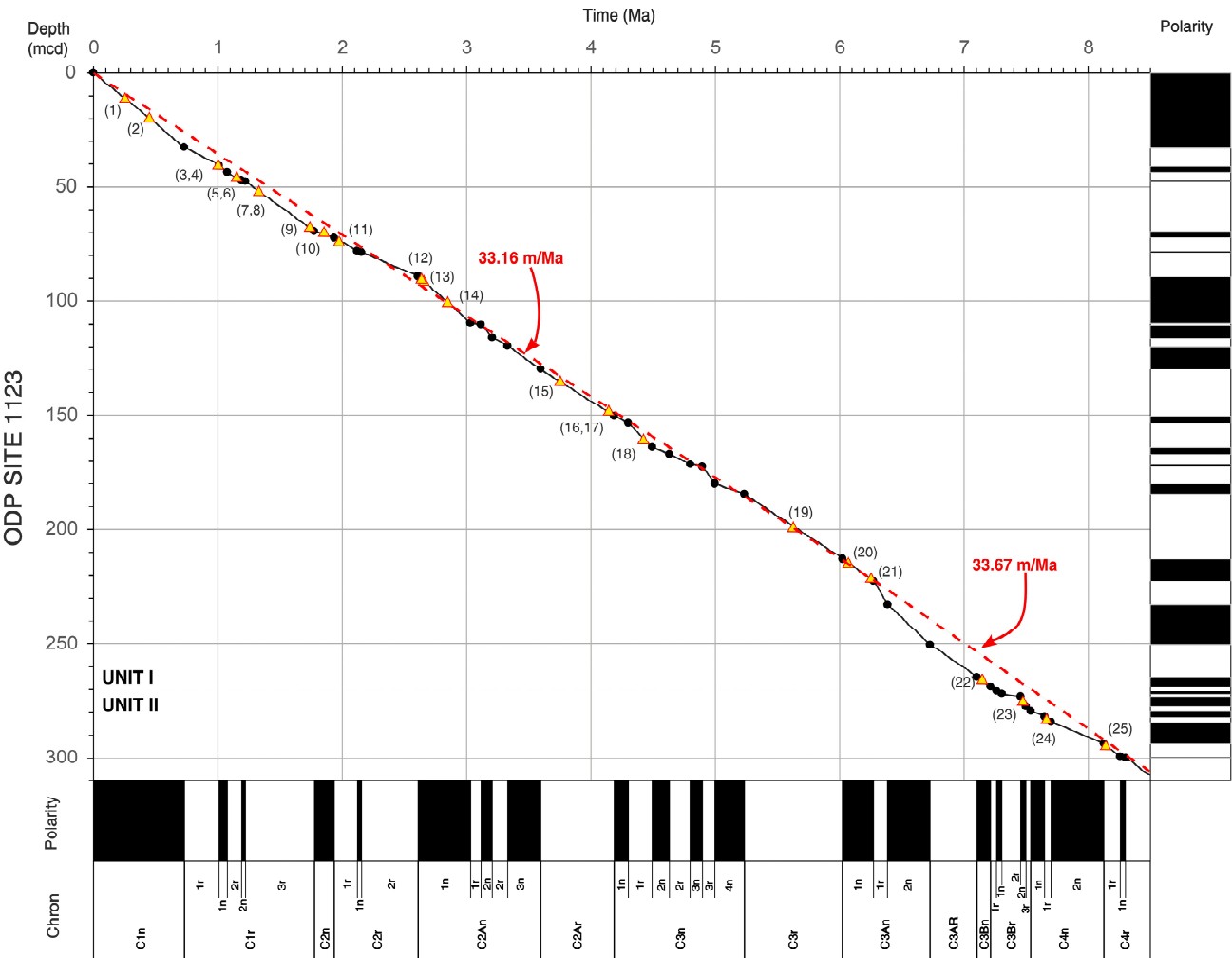

**Figure 5.** Cross-plot of magnetostratigraphic record at ODP Site 1123 and ages (Ma) recalibrated according to the ATS2020 [18] for the interval 0–300 mcd (Chron C1n to C4r). The black circles indicate the position of the polarity inversions while the yellow triangles indicate the position of the nannofossil events recognized along the section, numbered according to Figure 6.

The average sedimentation rate for lithostratigraphic Units I and II (Figure 2) is respectively 33.4 and 34.9 m/Myr and remarkably uniform (Figures 4 and 5). Possible small hiatuses or condensed intervals of less than 100 kyr are reported at about 270 and 340 mcd (age model in [1]). A major inflexion point at ~465 mcd denoted a change in sedimentation, separating the relatively fast deposition of lithostratigraphic Units I and II from the much slower Unit III where the sedimentation rate decreases to ~11.3 m/Myr. The lowermost portion of Unit III is characterized by an increased sedimentation rate (~32 m/Myr), probably due to the presence of a debris flow.

In the present study, the magnetostratigraphy and the calcareous nannofossils biostratigraphy at ODP Site 1123 were integrated and the numerical ages for each recognized bioevent were obtained through the linear interpolation between magnetic reversals.

Forty-three datum levels have been recognized. Some of them appeared as marker boundaries within the available nannofossil zonal schemes (Figure 3). Other levels have shown their potentiality in stratigraphic correlation and have been easily detected at Site ODP 1123, thus confirming their biostratigraphic usefulness.

The depth datum levels were calculated as mid-points within two subsequent samples and the corresponding age estimates are shown in Figure 6. In Table 1, we provided a comparison of the ages obtained in the present study with the ages reported for the same horizon in the previous literature. An estimate of the degree of reliability is also given. The identified bioevents and the inferred ages are discussed in the following sections.

| bioevent | marker species | Sample (older point) | | | | | | mcd | Ma | Sample (younger point) | | | | | mcd | Ma | ± m | mcd mid-point | Ma mid-point |
|---|---|---|---|---|---|---|---|---|---|---|---|---|---|---|---|---|---|---|---|
| 1 B | E. huxleyi | 1123 | B | 2 | H | 7 | 10 | 11.78 | 0.264 | 1123 | 2 | H | 6 | 110 | 11.28 | 0.253 | 0.25 | 11.53 | 0.26 |
| 2 T | P. lacunosa | 1123 | B | 3 | H | 4 | 105 | 20.17 | 0.452 | 1123 | 3 | H | 4 | 30 | 19.42 | 0.436 | 0.38 | 19.80 | 0.44 |
| 3 PT | medium Gephyrocapsa | 1123 | B | 5 | H | 3 | 30 | 40.82 | 1.015 | 1123 | 5 | H | 2 | 105 | 40.07 | 0.992 | 0.38 | 40.45 | 1.00 |
| 4 B | G. omega | 1123 | B | 5 | H | 3 | 30 | 40.82 | 1.015 | 1123 | 5 | H | 2 | 105 | 40.07 | 0.992 | 0.38 | 40.45 | 1.00 |
| 5 T | H. sellii | 1123 | B | 6 | H | 1 | 105 | 46.77 | 1.182 | 1123 | 5 | H | 5 | 30 | 43.82 | 1.086 | 1.48 | 45.30 | 1.13 |
| 6 T | C. macintyrei | 1123 | B | 6 | H | 1 | 105 | 46.77 | 1.182 | 1123 | 5 | H | 5 | 30 | 43.82 | 1.086 | 1.48 | 45.30 | 1.13 |
| 7 PB | medium Gephyrocapsa | 1123 | B | 6 | H | 4 | 105 | 51.27 | 1.317 | 1123 | 6 | H | 4 | 30 | 50.52 | 1.298 | 0.38 | 50.90 | 1.31 |
| 8 T | large Gephyrocapsa | 1123 | B | 6 | H | 4 | 105 | 51.27 | 1.317 | 1123 | 6 | H | 4 | 30 | 50.52 | 1.298 | 0.38 | 50.90 | 1.31 |
| 9 B | large Gephyrocapsa | 1123 | B | 8 | H | 1 | 105 | 67.27 | 1.725 | 1123 | 8 | H | 1 | 30 | 66.52 | 1.706 | 0.38 | 66.90 | 1.72 |
| 10 B | medium Gephyrocapsa | 1123 | B | 8 | H | 4 | 30 | 71.02 | 1.874 | 1123 | 8 | H | 3 | 105 | 70.27 | 1.832 | 0.38 | 70.65 | 1.85 |
| 11 T | D. brouweri | 1123 | B | 8 | H | 5 | 105 | 73.27 | 1.971 | 1123 | 8 | H | 5 | 30 | 72.52 | 1.948 | 0.38 | 72.90 | 1.96 |
| 12 T | D. pentaradiatus | 1123 | B | 10 | H | 2 | 105 | 89.49 | 2.619 | 1123 | 10 | H | 2 | 30 | 88.74 | 2.597 | 0.38 | 89.12 | 2.61 |
| 13 T | D. surculus | 1123 | B | 10 | H | 3 | 105 | 90.99 | 2.650 | 1123 | 10 | H | 3 | 30 | 90.24 | 2.635 | 0.38 | 90.62 | 2.64 |
| 14 T | D. tamalis | 1123 | B | 11 | H | 3 | 105 | 100.77 | 2.852 | 1123 | 11 | H | 3 | 30 | 100.02 | 2.836 | 0.38 | 100.40 | 2.84 |
| 15 T | R. pseudoumbilicus | 1123 | B | 14 | H | 5 | 30 | 135.02 | 3.749 | 1123 | 14 | H | 4 | 105 | 134.27 | 3.727 | 0.38 | 134.65 | 3.74 |
| 16 Bc | D. asymmetricus | 1123 | B | 15 | H | 7 | 31 | 148.87 | 4.149 | 1123 | 15 | H | 6 | 105 | 148.11 | 4.127 | 0.38 | 148.49 | 4.14 |
| 17 B | P. lacunosa | 1123 | B | 15 | H | 7 | 31 | 148.87 | 4.149 | 1123 | 15 | H | 6 | 105 | 148.11 | 4.127 | 0.38 | 148.49 | 4.14 |
| 18 Bc | H. sellii | 1123 | B | 16 | H | 6 | 105 | 159.69 | 4.418 | 1123 | 16 | H | 6 | 30 | 158.94 | 4.404 | 0.38 | 159.32 | 4.41 |
| 19 T | D. quinqueramus | 1123 | B | 21 | X | 2 | 30 | 198.36 | 5.622 | 1123 | 21 | X | 1 | 105 | 197.61 | 5.601 | 0.38 | 197.99 | 5.61 |
| 20 T | N. amplificus | 1123 | B | 22 | X | 6 | 105 | 214.88 | 6.076 | 1123 | 22 | X | 6 | 30 | 214.13 | 6.056 | 0.38 | 214.51 | 6.07 |
| 21 B | N. amplificus | 1123 | B | 23 | X | 5 | 105 | 222.91 | 6.277 | 1123 | 23 | X | 5 | 30 | 222.16 | 6.264 | 0.38 | 222.54 | 6.27 |
| 22 B | A. delicatus | 1123 | B | 28 | X | 2 | 105 | 266.41 | 7.156 | 1123 | 28 | X | 2 | 30 | 265.66 | 7.137 | 0.38 | 266.04 | 7.15 |
| 23 B | A. primus | 1123 | B | 29 | X | 1 | 102 | 274.49 | 7.470 | 1123 | 29 | X | 1 | 28 | 273.74 | 7.462 | 0.37 | 274.11 | 7.47 |
| 24 T | M. convallis | 1123 | B | 30 | X | 1 | 30 | 283.36 | 7.683 | 1123 | 29 | | | cc | 283.15 | 7.678 | 0.10 | 283.26 | 7.68 |
| 25 B | D. quinqueramus | 1123 | B | 31 | X | 1 | 105 | 293.81 | 8.130 | 1123 | 31 | X | 1 | 30 | 293.06 | 8.101 | 0.38 | 293.44 | 8.12 |
| 26 Bc | D. pentaradiatus | 1123 | B | 34 | X | 5 | 123 | 328.79 | 9.463 | 1123 | 34 | X | 5 | 42 | 327.98 | 9.431 | 0.41 | 328.39 | 9.45 |
| 27 T | C. coalitus | 1123 | B | 36 | X | 2 | 105 | 343.11 | 9.854 | 1123 | 36 | X | 2 | 30 | 342.36 | 9.832 | 0.38 | 342.74 | 9.84 |
| 28 B | M. convallis | 1123 | B | 35 | X | 5 | 30 | 337.46 | 9.720 | 1123 | 35 | X | 4 | 107 | 336.73 | 9.707 | 0.36 | 337.10 | 9.71 |
| 29 Bc | D. bellus | 1123 | B | 38 | X | 2 | 105 | 362.31 | 10.480 | 1123 | 38 | X | 2 | 30 | 361.56 | 10.454 | 0.38 | 361.94 | 10.47 |
| 30 B | C. coalitus | 1123 | B | 39 | X | 2 | 30 | 371.16 | 10.781 | 1123 | 39 | X | 1 | 105 | 370.41 | 10.755 | 0.38 | 395.44 | 10.77 |
| 31 Tc | C. miopelagicus | 1123 | B | 40 | x | 4 | 30 | 383.86 | 11.130 | 1123 | 40 | X | 3 | 105 | 383.11 | 11.160 | 0.38 | 383.49 | 11.15 |
| 32 Tc | D. kugleri | 1123 | B | 40 | X | 6 | 105 | 387.61 | 11.310 | 1123 | 40 | X | 6 | 30 | 386.86 | 11.199 | 0.38 | 387.24 | 11.25 |
| 33 B | D. kugleri | 1123 | B | 42 | X | 3 | 30 | 401.66 | 11.917 | 1123 | 42 | X | 2 | 105 | 400.91 | 11.876 | 0.38 | 401.29 | 11.90 |
| 34 Tc | C. premacintyrei | 1123 | B | 44 | | | cc | 422.98 | 12.513 | 1123 | 44 | X | 4 | 30 | 422.46 | 12.502 | 0.26 | 422.72 | 12.51 |
| 35 T | S. heteromorphus | 1123 | B | 48 | | | cc | 460.21 | 13.536 | 1123 | 48 | X | 3 | 100 | 460.16 | 13.530 | 0.05 | 460.18 | 13.53 |
| 36 Tc | H. ampliaperta | 1123 | B | 49 | X | 2 | 30 | 467.26 | 13.917 | 1123 | 49 | X | 2 | 105 | 466.51 | 13.853 | 0.38 | 466.89 | 13.89 |
| 37 X | D. deflandrei/D.variab. | 1123 | B | 51 | X | 5 | 30 | 490.96 | 15.581 | 1123 | 51 | X | 4 | 105 | 490.21 | 15.584 | 0.38 | 490.59 | 15.58 |
| 38 Tc | S. disbelemnos | 1123 | C | 21 | X | 6 | 102 | 528.50 | 17.774 | 1123 | 21 | X | 6 | 25 | 527.73 | 17.741 | 0.38 | 528.12 | 17.76 |
| 39 B | S. heteromorphus | 1123 | C | 22 | X | 2 | 105 | 532.23 | 17.937 | 1123 | 22 | X | 2 | 35 | 531.53 | 17.906 | 0.35 | 531.88 | 17.92 |
| 40 Tc | S. belemnos | 1123 | C | 22 | X | 6 | 23 | 537.41 | 18.211 | 1123 | 22 | X | 5 | 105 | 536.73 | 18.172 | 0.34 | 537.07 | 18.19 |
| 41 Bc | S. belemnos | 1123 | C | 24 | X | 1 | 30 | 549.84 | 18.729 | 1123 | 23 | | | cc | 548.99 | 18.723 | 0.09 | 549.09 | 18.73 |
| 42 T | T. carinatus | 1123 | C | 26 | X | 2 | 105 | 570.53 | 19.305 | 1123 | 26 | X | 2 | 30 | 569.78 | 19.285 | 0.38 | 570.16 | 19.29 |
| 43 AT | C. abisectus | 1123 | C | 28 | X | 1 | 105 | 588.43 | 19.790 | 1123 | 28 | X | 1 | 30 | 587.67 | 19.770 | 0.38 | 588.05 | 19.78 |
| **Marshall Paraconformity** | | 599.2 mcd (=19. 9 Ma) | | | | | | | | | | | | | | | | | |

**Figure 6.** Position of nannofossil events at ODP Site 1123 succession and inferred age according to the age models in Figures 5 and 7. For each horizon, the position (and relative age) of the two samples between which it falls are given as well of the mid-point.

*4.1. Holocene–Pleistocene (Holocene–Gelasian = Haweran–Nukumaruan)*

The most recent nannofossil events recognized at ODP Site 1123 are the first occurrence of *E. huxleyi*, between samples 1123B-2H-7-10 and 1123B-2H-6-110, and the disappearance of *P. lacunosa*, between samples 1123B-2H-7-10 and 1123B-2H-6-110.

The Pleistocene Epoch is finely subdivided by the distribution of gephyrocapsids, with the first occurrence of the medium *Gephyrocapsa* (which approximates the Calabrian GSSP [45]) detected between samples 1123B-8-H4-30 and 1123B-8-H3-105. The biostratigraphically useful distribution interval of large *Gephyrocapsa* [27,37] was detected between samples 1123B-8-H1-105 and 1123B-8-H1-30 (Base) and between samples 1123B-6-H4-105 and 1123B-6-H4-30 (top). The last was detected as coincident to a paracme interval of the medium *Gephyrocapsa*; above this horizon, the nannofossil assemblages are dominated

by small *Gephyrocapsa* specimens, up to the first occurrence of *Gephyrocapsa omega*, here detected between samples 1123B-5-H3-30 and 1123B-5-H2-105. This bioevent coincides with the end of the paracme interval of medium *Gephyrocapsa*, reported as a "reemG event" by [42].

Age estimates for bioevents from 1 to 10 (Table 1) compare well with the existing literature. Slight differences with the ages previously reported may be related to different methods of calibration (e.g., astrochronology) or to the different geomagnetic polarity time scales adopted [67].

Two further bioevents were reported as biostratigraphically useful in the present time interval, namely the last occurrences (T) of *H. sellii* and *C. macintyrei*, both recognized at the same level, between samples 1123B-6-H1-105 and 1123B-5-H5-30. The age estimation obtained for *H. sellii* T is slightly younger with respect to the previous calibration, thus revealing as moderately reliable, but the high degree of diachroneity characterizing the *C. macintyrei* T makes it an unreliable event.

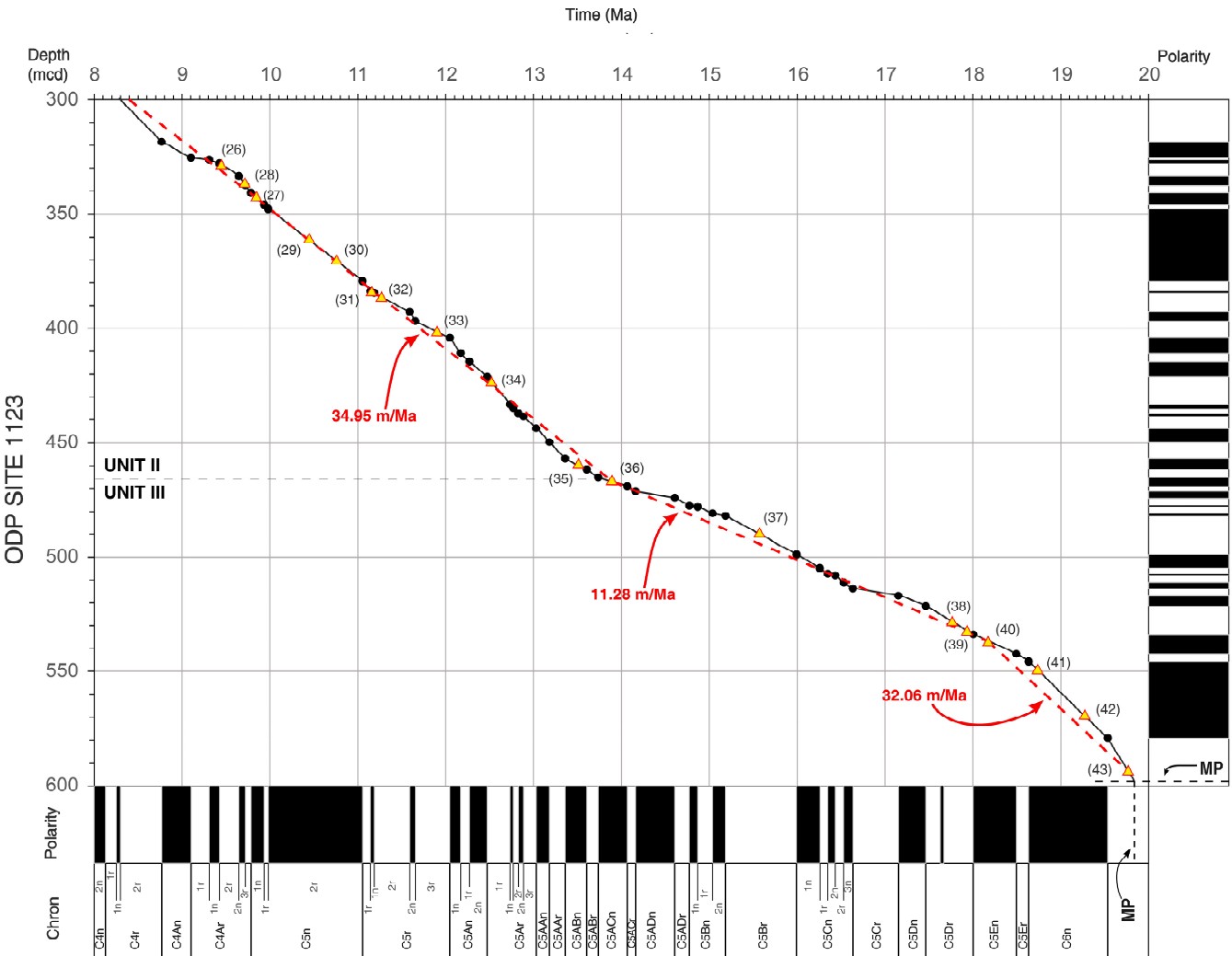

**Figure 7.** Cross-plot of magnetostratigraphic record at ODP Site 1123 and ages (Ma) recalibrated according to the ATS2020 [18] from 300 mcd to the Marshall Paraconformity (MP) at ~599 mcd (Chron C4r-C6). The black circles indicate the position of the polarity inversions while the yellow triangles indicate the position of the nannofossil events recognized along the section, numbered according to Figure 6.

**Table 1.** Comparison between the age estimates (expressed in Ma) obtained in the present study for the nannofossil bioevents detected at ODP Site 1123 and previous calibration for the same or similar horizon. The 1 stands for present study, 2 stands for Oceanic succession calibration and 3 stands for Mediterranean succession calibration.

| Epoch | Bioevent | | Marker Species | 1 | 2 | | 3 | Reliability Degree | Chronostratigraphic Remarks |
|---|---|---|---|---|---|---|---|---|---|
| Pleistocene-Holocene | 1 | B | *E. huxleyi* | **0.26** | 0.29 [27] | 0.24 [38] | 0.26 [39] | good | Base Late Pleistoc. 0.13 [30] |
| | 2 | T | *P. lacunosa* | **0.44** | 0.43 [29] | | 0.46 [40] | good | Chibanian GSSP-0.77 [41] |
| | 3 | PT | m *Gephyrocapsa* | **1.00** | 1.06 [42] | | | good | |
| | 4 | B | *G. omega* | **1.00** | | | 0.96 [40] | good | |
| | 5 | T | *H. sellii* | **1.13** | 1.24 [37] | | 1.26 [40] | moderate | |
| | 6 | T | *C. macintyrei* | **1.13** | 1.60 [29] | | 1.67 [43] | unreliable | |
| | 7 | PB | m *Gephyrocapsa* | **1.31** | | | | good | |
| | 8 | T | l *Gephyrocapsa* | **1.31** | 1.25 [37] | | 1.24 [43] | good | |
| | 9 | B | l *Gephyrocapsa* | **1.72** | 1.59 [42] | | 1.61 [43] | good | |
| | 10 | B | m *Gephyrocapsa* | **1.85** | 1.71 [42] | 1.69 [44] | 1.71 [43] | good | Calabrian GSSP-1.80 [45] |
| | 11 | T | *D. brouweri* | **1.96** | 1.93 [46] | 2.06 [44] ~1.93 [47] | 1.95 [43] | good | |
| | 12 | T | *D. pentaradiatus* | **2.61** | 2.39 [46] | | 2.51 [48] | good | Gelasian GSSP-2.58 [49] |
| Late Miocene-Pliocene | 13 | T | *D. surculus* | **2.64** | 2.53 [46] | ~2.37 [47] | 2.55 [48] | good | |
| | 14 | T | *D. tamalis* | **2.84** | 2.76 [46] | 2.87 [44] | 2.82 [40] | good | |
| | 15 | T | *R. pseudoumbilicus* | **3.74** | 3.82 [46] | ~3.75 [47] | 3.85 [50] | good | Piacenzian GSSP-3.60 [51] |
| | 16 | Bc | *D. asymmetricus* | **4.14** | 4.04 [29] | | 4.11 [50] | good | |
| | 17 | B | *P. lacunosa* | **4.14** | 4.00 [52] | | | good | |
| | 18 | Bc | *H. sellii* | **4.41** | | | 4.62 [53] | moderate | |
| | 19 | T | *D. quinqueramus* | **5.61** | 5.53 [29] | | | good | Zanclean GSSP-5.33 [54] |
| | 20 | T | *N. amplificus* | **6.07** | 5.98 [29] | | 5.85 [55] | unreliable | |
| | 21 | B | *N. amplificus* | **6.27** | 6.82 [29] | | 6.69 [55] 6.21 [56] | unreliable | |
| | 22 | B | *A. delicatus* | **7.15** | | | 7.13 [55] | good | Messinian GSSP-7.24 [57] |
| | 23 | B | *A. primus* | **7.47** | 7.45 [58] | 7.39 [29] | 7.42 [55] | good | |
| | 24 | T | *M. convallis* | **7.68** | 7.78 [59] | | 7.78 [55] | moderate | |
| | 25 | B | *D. quinqueramus* | **8.12** | 8.10 [44] | | | good | |
| | 26 | Bc | *D. pentaradiatus* | **9.45** | | | 9.37 [55] | moderate | |
| | 27 | T | *C. coalitus* | **9.84** | 9.62 [58] | 9.65 [29] | | unreliable | |
| | 28 | B | *M. convallis* | **9.71** | 9.75 [29] | | 9.23 [55] | moderate | |
| | 29 | Bc | *D. bellus* | **10.47** | 10.64 [58] | | 10.38 [59] | moderate | |
| | 30 | B | *C. coalitus* | **10.77** | 10.89 [58] | 10.79 [29] | | unreliable | |
| | 31 | Tc | *C. miopelagicus* | **11.15** | 11.04 [58] | 10.61 [29] | 10.90 [59] | moderate | |
| | 32 | Tc | *D. kugleri* | **11.25** | 11.61 [58] | 11.60 [29] | 11.60 [59] | unreliable | Tortonian GSSP-11.61 [60] |
| Early-Middle Miocene | 33 | B | *D. kugleri* | **11.90** | 11.89 [58] | 11.88 [29] | 11.90 [59] | unreliable | |
| | 34 | Tc | *C. premacintyrei* | **12.51** | 12.57 [29] | | 12.51 [61] | good | |
| | 35 | T | *S. heteromorphus* | **13.53** | 13.60 [58] | | 13.59 [62] | good | Serravallian GSSP-13.82 [63] |
| | 36 | T | *H. ampliaperta* | **13.89** | 14.86 [29] | | | unreliable | |
| | 37 | X | *D.deflandrei/D.variab.* | **15.58** | 15.80 [29] | | | moderate | Langhian GSSP-15.99 [64] |
| | 38 | T | *S. disbelemnos* | **17.76** | | | 17.69 [65] | to be tested | |
| | 39 | B | *S. heteromorphus* | **17.92** | 17.65 Bc [29] | | 17.99 [65] | good | |
| | 40 | Tc | *S. belemnos* | **18.19** | 17.94 T [29] | | 18.02 T [65] | moderate | |
| | 41 | Bc | *S. belemnos* | **18.73** | 19.01 B [29] | | 19.12 B [65] | moderate | Base Burdigalian Stage? [66] |
| | 42 | T | *T. carinatus* | **19.29** | 19.18 [29] | | | moderate | |
| | 43 | AT | *C. abisectus* | **19.78** | | | | to be tested | |

From a chronostratigraphic point of view, we could observe that: (i) the base of the Late Pleistocene [30] falls between the base of *E. huxleyi* and the top of *P. lacunosa*; (ii) the base of the Middle Pleistocene (Chibanian GSSP at 0.77 Ma, [41]) falls between the top of *P. lacunosa* and the base of *G. omega* (coincident with the top paracme of the medium *Gephyrocapsa*); (iii) the Calabrian GSSP (1.80 Ma at Vrica section, [45]) is approximated by the base of the medium *Gephyrocapsa*.

Two more useful bioevents based on discoasterids were detected: (i) the disappearance (T) of *D. brouweri*, which represent the last stock of *Discoaster* before they became extinct (between samples 1123B-8-H5-105 and 1123B-8-H5-30). The estimated age of this bioevent at ODP Site 1123 fits well with the previous calibrations; (ii) the disappearance of *D. pentaradiatus* (between samples 1123B-10-H2-105 and 1123B-10-H2-30) that approximates the Gelasian GSSP (base of the Pleistocene epoch (2.58 Ma at Mt. St. Nicola Section [48,68]).

### 4.2. Pliocene–Late Miocene (Piacenzian–Tortonian = Mangapanian–Tongaporutuan)

The Pliocene epoch was adequately subdivided by the distribution of several species belonging to the *Discoaster* genus, which were well represented and preserved in this time interval at ODP Site 1123. We easily detected the top of *D. surculus* (between samples 1123B-10-H3-105 and 1123B-10-H3-30) and *D. tamalis* (between samples 1123B-11-H3-105 and 1123B-11-H3-30). *Discoaster asymmetricus* was sporadically present in low frequencies from core 33, but the common presence of the species occurred between samples 1123B-15-H7-31 and 1123B-15-H6-105, together with the first occurrence (B) of *P. lacunosa*.

The top of *R. pseudoumbilicus* between samples 1123B-14H-5-30 and 1123B-14H-4-105 and the common presence (Bc) of *H. sellii* between samples 1123B-16H-6-105 and 1123B-16H-6-30 provided useful tools for subdividing the Early Pliocene; indeed, taxa such as ceratholitids (e.g., *C. rugosus* and *C. acutus*) traditionally used as markers of this time interval are very rare or absent at ODP Site 1123. The disappearance of *Sphenolithus* spp. was not detected as low frequencies of this taxa are also present in the late Pliocene and Pleistocene, probably due to reworking.

The top of *Discoaster quinqueramus* between samples 1123B-21X-2-30 and 1123B-21X-1-105 indicated that we are very close to the Pliocene/Miocene boundary [18].

Pliocene bioevents are mainly based on *Discoaster* species, and the estimated ages at ODP Site 1123 well fit with previous calibrations (Table 1). We also provide ages for (i) the top of *R. pseudoumbilicus*, which approximates the Piacenzian GSSP [51]; (ii) the base of *P. lacunose*, that fits with the calibration obtained by [52] in central Indian Ocean and (iii) the Bc of *H. sellii*, slightly younger than the calibration obtained in the Mediterranean [53,69].

The Late Miocene can be subdivided based upon a series of fast-evolving taxa such as *Amaurolithus* and *Catinaster*. Although specimens of the *Amaurolithus* genus are discontinuously distributed within the ODP Site 1123 succession, we were able to detect the B of *A. primus* between samples 1123B-29X-1-102 and 1123B-29X-1-28, and the B of *A. delicatus* between samples 1123B-28X-2-105 and 1123B-28X-2-30. The ages obtained for these two bioevents were comparable with those available from the existing literature, and they fall between the Tortonian/Messinian boundary [57].

On the other hand, *N. amplificus*, considered a useful marker in the late Miocene, was discontinuously recorded in very few specimens. The base and the top of this species were detected between samples 1123B-23X-5-105 and 1123B-23X-5-30 and between samples 1123B-22X-6-105 and 1123B-22X-6-30, respectively. Nevertheless, none of the two bioevents can be considered reliable, as testified by the diachronous ages obtained with respect to previous calibrations already observed in Mediterranean sections [56,70].

*Catinaster coalitus* is discontinuously distributed in low frequencies from samples 1123B-36X-2-105 to 1123B-39X-2-30 and very rare specimens of *C. calyculus* occurred within the same interval, thus events based on *Catinaster* cannot be considered as reliable at ODP Site 1123.

Bad preservation of discoasterids was often aggravated by overgrowth and hampered the recognition of species such as *D. hamatus*, *D. neohamatus*, *D. loeblichii* and *D. berggrenii*.

Nevertheless, we were able to detect some additional reliable bioevents based on the estimate ages and used them for stratigraphic correlations, namely: (i) Bc of *D. pentaradiatus* between samples 1123B-34X-5-123 and 1123B-34X-5-42; (ii) Bc of *D. bellus* between samples 1123B-38X-3-105 and 1123B-38X-2-30 and B of *D. quinqueramus* between samples 1123B-31X-1-105 and 1123B-31X-1-30. *Discoaster surculus* is very discontinuous in the lower part of its distribution range, thus the Bc of the species was impossible to be established.

Further bioevents characterizing this time interval are the restricted distribution range of *M. convallis*, with the top found between samples 1123B-30X-1-30 and 1123B-29X-CC; the base was detected between samples 1123B-35X-5-30 and 1123B-35X-4-107. These bioevents ranged from 9.71 to 7.68 Ma, according to our results. The last common occurrence (Tc) of *C. miopelagicus* was recorded between samples 1123B-40X-4-30 and 1123B-40X-3-105, and the estimated age of this bioevent is in good agreement with the one reported by [58].

The short distribution range of *D. kugleri* (top between samples 1123B-40X-6-105 and 1123B-40x-6-30; base between samples 1123B-42X-3-30 and 1123B-42x-2-105) characterizes the Tortonian–Serravallian transition [59], but this species is very rare at ODP Site 1123. Thus, in spite of the good agreement of the inferred age for the base of this species, with respect to previous calibrations, neither the base nor the top can be considered reliable bioevents, even if the presence of *D. kugleri* may help identify the Tortonian GSSP [60].

### 4.3. Middle–Early Miocene (Serravallian–Burdigalian = Waiauan–Otaian)

Tc of *C. premacintyrei*, which proved to be a useful tool for stratigraphic correlation, occurred between samples 1123B-44X-cc and 1123B-44x-4-30. The age obtained for this bioevent is comparable with other calibrations, thus confirming its reliability.

T of *S. heteromorphus* was found between samples 1123B-48-cc and 1123b-48X-3-100, with an estimated age of 13.53 Ma. This event fits well with previous determinations and is the best nannofossil event for approximating the Serravallian GSSP [63].

In this interval, most specimens of *Discoaster* spp. are overgrown due to re-crystallization, and the recognition of species such as *D. signus* was impossible. In this condition, we only could observe the crossover (X) between *Discoaster deflandrei* and *D. variabilis*, which was detected between samples 1123B-49X-3-105 and 1123B-49X-3-29. The resulting age of 15.58 Ma is slightly younger than that calibrated by [29], but the bioevent can be considered moderately reliable and useful for approximating the Langhian GSSP [64].

T of *H. ampliaperta* is historically considered a biostratigraphically useful taxon (Table 1), but it was difficult to detect because the species was very rare and discontinuous at ODP Site 1123. Last specimens occurred between samples 1123B-49X-2-30 and 1123B-49X-2-105, but the resulting age of 13.89 Ma is about 1 Myr younger than that reported by previous calibrations.

In Early Miocene, a few significant bioevents were recognized based upon the *Sphenolithus* genus, namely the B of *S. heteromorphus* (between sample 1123C-22X-2-105 and 1123C-22X-2-35) and the Bc and Tc of *S. belemnos*, respectively, between sample 1123C-22X-6-23 and 1123C-22X-5-105 and between 1123C-24X-1-30 and 1123C-23-cc.

The estimated age of 17.92 Ma for the B of *S. heteromorphus* at ODP Site 1123 fit well with the one obtained by [65] in the Mediterranean, and is older than the one obtained by [29], which did however consider the Bc of the species.

The ages obtained for Tc and Bc of *S. belemnos* are partially in agreement with previous calibrations which nonetheless consider the T and the B of the species. The quality of these events could also be distorted by the presence of a debris flow that affects this part of the ODP Site 1123 succession.

It is worth mentioning that the base of *S. belemnos* represents one of the criteria used to recognize the base of the Burdigalian Stage, which still remains inconclusive [18,66].

*Sphenolithus disbelemnos* is widely known as a useful tool in biostratigraphy [28,65,71,72] and has been recorded since the base of the succession above the Marshall Paraconformity. The Tc of the species was found between 1123C-21X-6-102 and 1123C-21X-6-25 and

had an age estimate of 17.76 Ma, comparable with that obtained in some Mediterranean sections [65,69].

The Top of *T. carinatus* (between 1123C-26X-2-30 and 1123C-26X-2-30) had an age of 19.29 Ma, comparable with the calibration by [29], but it can be considered only moderately reliable due to the scarce and discontinuous presence of the species.

The oldest nannofossil event recognized above the Marshall Paraconformity was a sharp drop in the abundance of large (>10 μm according to Rio et al., 1990 [27]) *C. abisectus* (between samples 1123C-28X-1-105 and 1123C-28X-1-30), previously observed in other Pacific successions [33,73] and considered a useful tool for detecting the basal Miocene, but no previous age estimates are available. The age obtained for this bioevent at ODP Site 1123 is 19.78 Ma and can be considered moderately reliable pending testing in other successions.

Below the Marshall Paraconformity, at 599.20 mcd, Early Oligocene nannofossil assemblages occurred. Based upon the calculated sedimentation rate, the age of the oldest sediments immediately above the paraconformity was estimated to be 19.9 Ma.

## 5. Conclusive Remarks

The quantitative analysis of the calcareous nannofossil content of ODP Site 1123 allowed a refined biostratigraphic subdivision via the recognition of 43 bioevents, whose numerical ages were estimated through a comparison with the excellent magnetostratigraphic record available for the same succession. An evaluation of the reliability of the detected bioevents was also given, through a comparison with previous age calibrations.

We were able to detect many of the bioevents used in previous nannofossil zonal schemes but were also to verify the absence or scarce usefulness of well-established stratigraphic markers such as the *Ceratholithus* and *Catinaster* genera and some species of the *Discoaster* genus (e.g., *D. berggrenii*, *D. hamatus*, *D. loeblichii*, *D. neorectus* and *D. signus*). In addition, the distribution of taxa traditionally considered as biostratigraphically useful, such as with *N. amplificus* and *H. ampliaperta*, was revealed to be poorly applicable to ODP Site 1123 succession. On the other hand, we were able to highlight additional bioevents which confirmed their potentiality for stratigraphic correlations and highly improved the bio- and chronostratigraphic resolution in the last 20 Myr in the Southern Ocean region.

The obtained average bio- and chronostratigraphic resolution, only considering the bioevents characterized by an acceptable degree of reliability, is of about 0.6 Myr along the whole section, which increases to about 0.3 in the Pliocene–Holocene time interval.

However, some uncertainties still remain, such as the low bio- and chronostratigraphic resolution that characterizes the upper part of the Burdigalian and the lower Langhian stages, as well as the need to test the validity of some additional bioevents (e.g., *S. disbelemnos*; *C. abisectus*) that should deserve further investigation.

**Author Contributions:** Conceptualization, A.D.S. and V.B.; methodology and material, A.D.S.; biostratigraphic analysis, A.D.S., N.B., V.B. and N.M.D.; litostratigraphy and age model, S.D. and S.U.; data elaboration, A.D.S., V.B. and L.B.; writing—original draft preparation, A.D.S.; writing—review and editing, L.B. and N.B.; figures, S.D. and S.U.; supervision and funding acquisition, A.D.S. All authors have read and agreed to the published version of the manuscript.

**Funding:** This research received no external funding.

**Institutional Review Board Statement:** Not applicable.

**Informed Consent Statement:** Not applicable.

**Data Availability Statement:** Not applicable.

**Acknowledgments:** The authors wish to thank the Special Issue editors, Giulia Bosi (JMSE Academic Editor) and the anonymous reviewers for their constructive comments that helped to significantly improve the manuscript. The technician Alfio Viola is warmly thanked for their help with the SEM microphotography (instrument: Tescan Vega\\LMU at Department of Biological, Geological and Environmental Science, University of Catania).

**Conflicts of Interest:** The authors declare no conflict of interest.

## Appendix A. Figures of Nannofossils Index Markers at ODP Site 1123

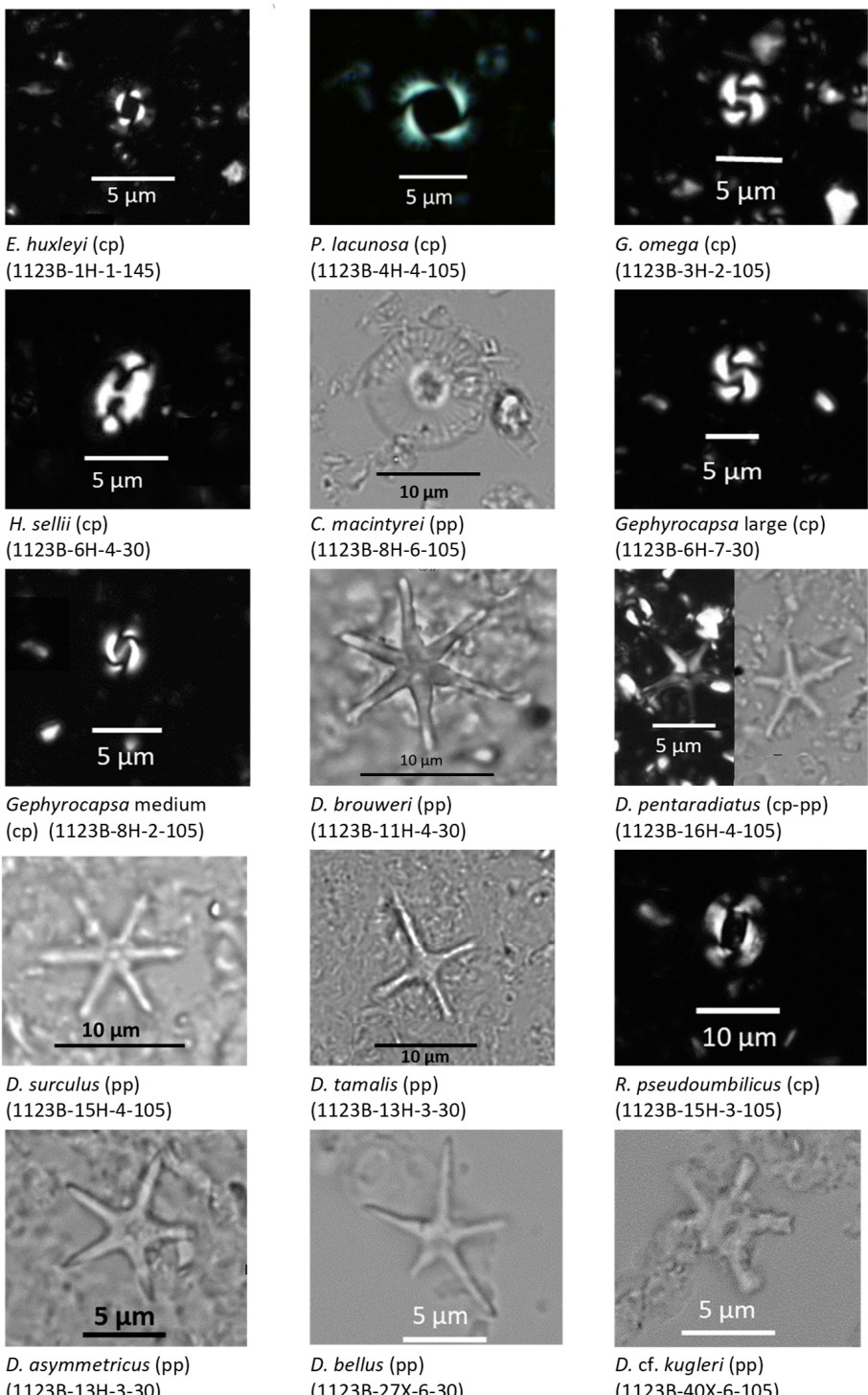

**Figure A1.** Micro-photographs of Holocene–Late Miocene nannofossil index markers at ODP Site 1123 (Zeiss Axioscope polarizing microscope at 1000–1600× magnification; cross-polarized = cp and plane-polarized = pp light).

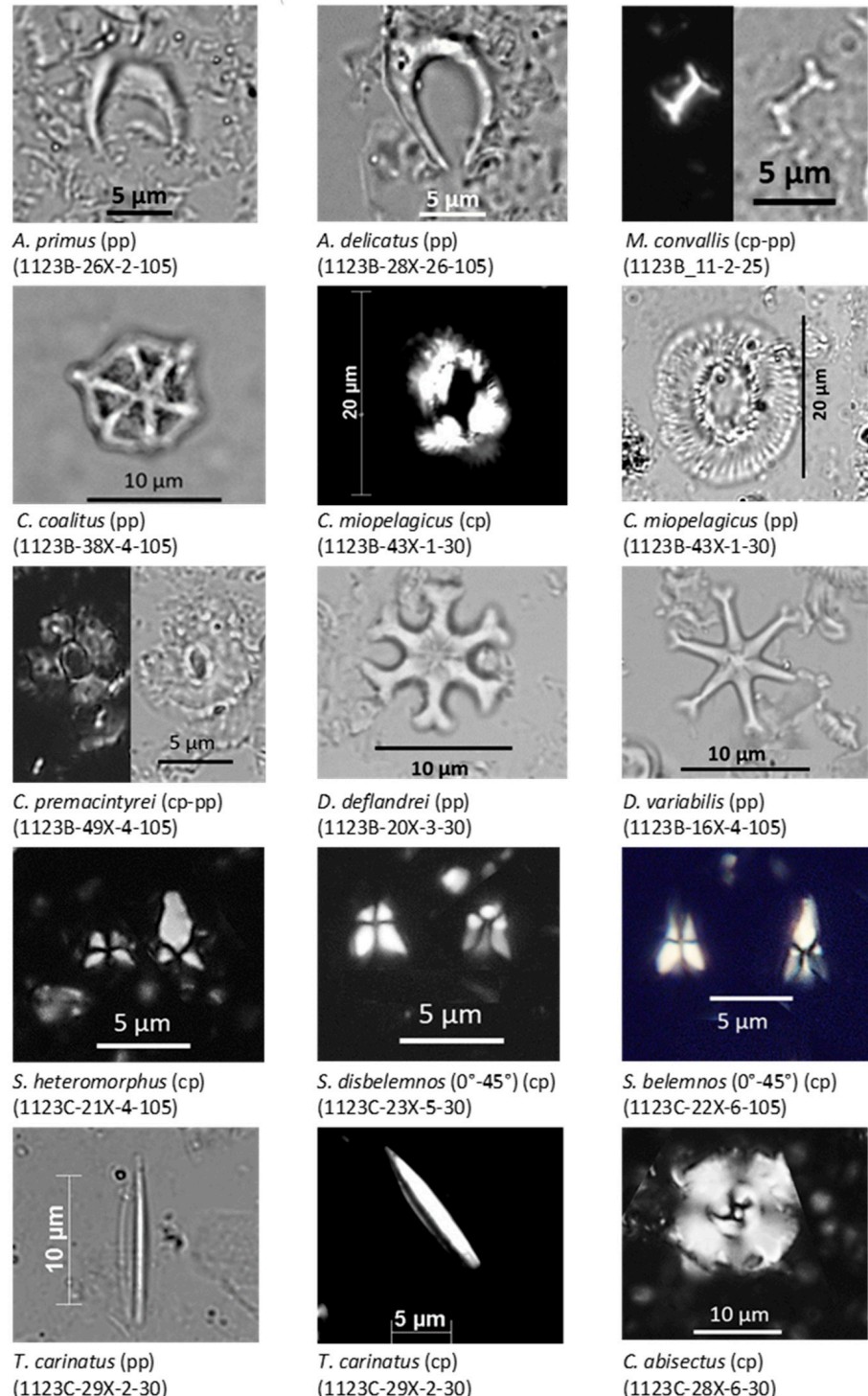

**Figure A2.** Micro-photographs of Late–Early Miocene nannofossil index markers at ODP Site 1123 (Zeiss Axioscope polarizing microscope at 1000–1600× magnification; cross-polarized = cp and plane-polarized = pp light).

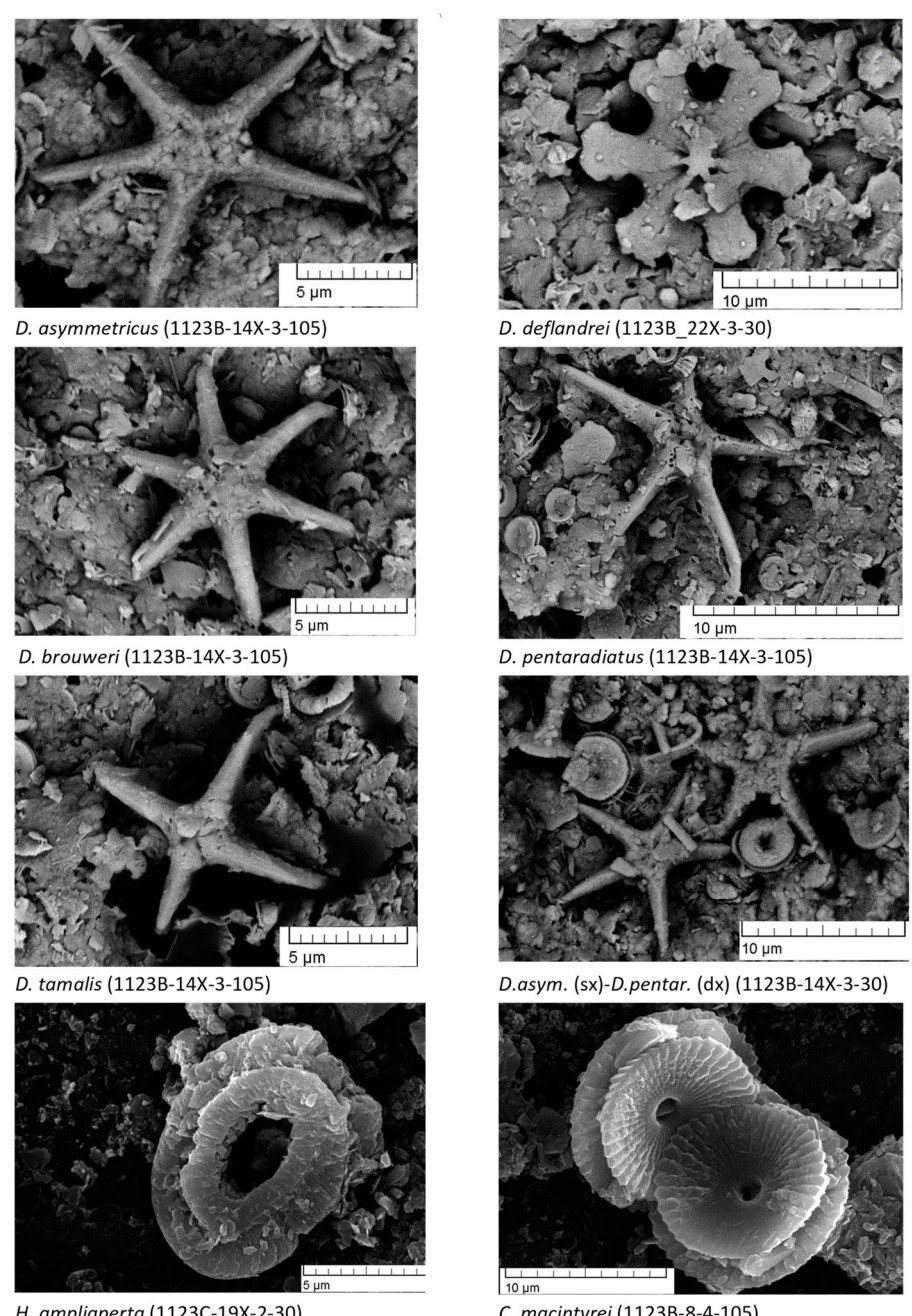

**Figure A3.** SEM micro-photographs of selected calcareous nannofossils at ODP Site 1123 (Scanned Electron Microscope Tescan Vega\\LMU).

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
