# Peer review of "Calcareous Nannofossil Biostratigraphy and Biochronology at ODP Site 1123 (Offshore New Zealand): A Reference Section for the Last 20 Myr in the Southern Ocean"

_jmse, doi:10.3390/jmse11020408_

Round 1

Reviewer 1 Report

Dear Authors,

Attached please find a manuscript I have reviewed with corrections I propose.

It would be very useful if the you add two or three plates with illustrations of marker species.

I propose publication of this manuscript after minor changes.

Author Response

Reviewer 1 Comments and Suggestions for Authors

Dear Authors,

Attached please find a manuscript I have reviewed with corrections I propose. It would be very useful if the you add two or three plates with illustrations of marker species. I propose publication of this manuscript after minor changes.

Dear Reviewer 1, thank you for your revision.

We observed your comments given in the text and modified it according to your suggestions.
In addition, following up on your suggestion, we included 3 plates illustrating the nannofossil markers.

Reviewer 2 Report

The Miocene to Quaternary Calcareous nannofossil biostratigraphy and biochronology at ODP Site 1123 are well represents in the manuscript, which is very valuable for the global Calcareous nannofossil biostratigraphy and biochronology, as well as geological events during the Miocene to Quaternary periods. I suggest to publish after minor modifications.

Minor modifications and suggestions

1.     In the first paragraph of the article, the significance of the research (including ODP Site 1123) should be emphasized.

2.     Figure 1,Is there any Holocene strata in the ODP 1123 borehole?

3.     Figure 4. Chronostratigraphic framework adopted in the present study. The figure should put in the central part.

4.     3. Neogene-Quaternary Calcareous Nannofossils biostratigraphy: State of Artand3. Results: Calcareous Nannofossils Biostratigraphy and Biochronology at ODP Site 1581123. I guess the later is “4”. However, for the latter, I suggest to delete “Results”, or should be changed to change to “Results and Discussions”.

5.     “4.1 Holocene-Pleistocene”, I suggest to change to “Pleistocene- Holocene”, and the sane for others.

6.     If a figure for typical photos of Calcareous nannofossil, it will be much better.

Author Response

Dear Reviewer 2, thank you for your revision.

Here you find a point-by-point response to your comments:

1. In the first paragraph of the article, the significance of the research (including ODP Site 1123) should be emphasized. We agree. This point was emphasized in the Introduction section. 

2. Figure 1,Is there any Holocene strata in the ODP 1123 borehole?Correct. We amended the figure accordingly.

3. Figure 4. Chronostratigraphic framework adopted in the present study. The figure should put in the central part. OK

4. 3. Neogene-Quaternary Calcareous Nannofossils biostratigraphy: State of Art”and“ 3. Results: Calcareous Nannofossils Biostratigraphy and Biochronology at ODP Site 1123”. I guess the later is “4”. Correct.

However, for the latter, I suggest to delete “Results”, or should be changed to change to “Results and Discussions”. Correct, we deleted "Results:"

5. “4.1 Holocene-Pleistocene”, I suggest to change to “Pleistocene- Holocene”, and the sane for others. We preferred to maintain the stratigraphic order of the cors, even in the description of bioevents.

6. If a figure for typical photos of Calcareous nannofossil, it will be much better. Following up on your suggestion, we included 3 plates illustrating nannofossil markers

Reviewer 3 Report

This paper presents a well documented record of calcareous nannofossils from the Southern Ocean. The paper concise and well written, all the data are well documented and the overall paper is easy to read. The interpretations are well sustained by the daya.

I annotated some very minor corrections:

Row 40: and the succession also is accompanied by an excellent >> and with an excellent

Row 87: follows >> follow

Row 129: component >> components

Row 179: small squares >> black circles;  small triangles >> yellow triangles; the red arrows indicate the average sedimentation rate for the interval indicated by the dashed red line.

Same comment for the next figure. It could be useful to indicate in these figures the extent of each lithostratigraphic unit, so that it can be better visualized when commenting on sedimentation rates in the text. By the way, it is indicated as 33.91 m/My in the text but different values are on the figures.

Row 188: 33,91 >> 33.91

Row 193: reduces >> decreases

Row 239: with respect the o previous >> with respect to the previous

Row 251: well fit >> well fits

Author Response

Dear Reviewer 3, thank you for your revision.

Here you find a point-by-point response to your comments:

Row 40: and the succession also is accompanied by an excellent >> and with an excellent OK

Row 87: follows >> follow OK

Row 129: component >> components OK

Row 179: small squares >> black circles; small triangles >> yellow triangles; the red arrows indicate the average sedimentation rate for the interval indicated by the dashed red line.

Same comment for the next figure. It could be useful to indicate in these figures the extent of each lithostratigraphic unit, so that it can be better visualized when commenting on sedimentation rates in the text. By the way, it is indicated as 33.91 m/My in the text but different values are on the figures.

Thank you indeed, we amended the figures accordingly

Row 188: 33,91 >> 33.91 OK

Row 193: reduces >> decreases OK

Row 239: with respect the o previous >> with respect to the previous OK

Row 251: well fit >> well fits OK